# Induction of Apoptosis via Inactivating PI3K/AKT Pathway in Colorectal Cancer Cells Using Aged Chinese Hakka Stir-Fried Green Tea Extract

**DOI:** 10.3390/molecules27238272

**Published:** 2022-11-27

**Authors:** Xinyue Zhang, Haiying Huang, Shili Sun, Dongli Li, Lingli Sun, Qiuhua Li, Ruohong Chen, Xingfei Lai, Zhenbiao Zhang, Xi Zheng, Wing-Leung Wong, Shuai Wen

**Affiliations:** 1School of Biotechnology and Health Sciences, Wuyi University, Jiangmen 529020, China; 2Tea Research Institute, Guangdong Key Laboratory of Tea Resources Innovation & Utilization/Guangdong Academy of Agricultural Sciences, Guangzhou 510640, China; 3Tea Research Institute, Meizhou Academy of Agriculture and Forestry Sciences, Meizhou 514071, China; 4International Healthcare Innovation Institute (Jiangmen), Jiangmen 529040, China

**Keywords:** cell apoptosis, cell-cycle arrest, PI3K/AKT signalling, green tea extract, effect of aging time

## Abstract

Food extract supplements, with high functional activity and low side effects, play a recognized role in the adjunctive therapy of human colorectal cancer. The present study reported a new functional beverage, which is a type of Chinese Hakka stir-fried green tea (HSGT) aged for several years. The extracts of the lyophilized powder of five HSGT samples with different aging periods were analyzed with high-performance liquid chromatography. The major components of the extract were found to include polyphenols, catechins, amino acids, catechins, gallic acid and caffeine. The tea extracts were also investigated for their therapeutic activity against human colorectal cancer cells, HT-29, an epithelial cell isolated from the primary tumor. The effect of different aging time of the tea on the anticancer potency was compared. Our results showed that, at the cellular level, all the extracts of the aged teas significantly inhibited the proliferation of HT-29 in a concentration-dependent manner. In particular, two samples prepared in 2015 (**15Y**, aged for 6 years) and 2019 (**19Y**, aged for 2 years) exhibited the highest inhibition rate for 48 h treatment (cell viability was 50% at 0.2 mg/mL). Further, all the aged tea extracts examined were able to enhance the apoptosis of HT-29 cells (apoptosis rate > 25%) and block the transition of G1/S phase (cell-cycle distribution (CSD) from <20% to >30%) population to G2/M phase (CSD from nearly 30% to nearly 10%) at 0.2 mg/mL for 24 h or 48 h. Western blotting results also showed that the tea extracts inhibited cyclin-dependent kinases 2/4 (CDK2, CDK4) and CylinB1 protein expression, as well as increased poly ADP-ribose polymerase (PRAP) expression and Bcl2-associated X (Bax)/B-cell lymphoma-2 (Bcl2) ratio. In addition, an upstream signal of one of the above proteins, phosphatidylinositol 3-kinase (PI3K)/protein kinase B (AKT) signalling, was found to be involved in the regulation, as evidenced by the inhibition of phosphorylated PI3K and AKT by the extracts of the aged tea. Therefore, our study reveals that traditional Chinese aged tea (HSGT) may inhibit colon cancer cell proliferation, cell-cycle progression and promoted apoptosis of colon cancer cells by inactivating PI3K/AKT signalling.

## 1. Introduction

Human colon cancer is a malignant tumor with extremely high morbidity and mortality worldwide. The disease thus seriously threatens human health. Although colon cancer has previously mostly occurred in developed countries, the incidence and mortality of this disease have gradually increased in developing countries, including China, probably due to the social development and lifestyle changes in recent decades [1,2]. According to the newly released data, the 5-year survival for patients with metastatic colon cancer is less than 20% and more than 50,000 patients are estimated to have died from colon cancer in 2020 [3]. Surgical resection and chemotherapy are conventional treatments for colon cancer, but postoperative recurrence and adverse drug side effects bring great pain and nightmares to patients [4]. Therefore, novel and effective therapeutic strategies with low side effects are essential to treat the disease and improve patient outcomes.

The uncontrolled growth of colon cancer cells depends on persistent activation of corresponding intracellular proliferative signals. Under the stimulation of various extracellular growth factors, receptor tyrosine kinases (RTKs) are activated to stimulate the subsequent increase in the phosphorylation and activity of phosphatidylinositol 3-kinase (PI3K). PIP3 is phosphorylated by PI3K, which binds to AKT and promotes phosphorylation at Thr308 of AKT [5,6]. Some recent studies have shown that the activation of the PI3K/AKT pathway plays a positive role in the carcinogenesis, cell survival, migration, and metabolism of colon cancer [7,8,9]. Intervention by inhibitors or RNA interference technology, or the PI3K/AKT pathway and related upstream and downstream sites, may block the pathway, inhibit cell proliferation and cycle, and promote cell apoptosis. For example, it was reported that berberine inhibited PI3K and AKT expression and induced apoptosis and cell-cycle arrest in SW480 cells [10]. Small interfering RNA and molecular inhibitors of the mismatch repair gene MutL Homolog 1 (MLH1) suppressed colon cancer sensitivity to cetuximab treatment via PI3K/AKT signalling [11]. As a downstream signal of PI3K/AKT, the progression of cell cycle is necessary for colon cancer proliferation. Several small-molecule inhibitors, such as botulin and periplocymarin, blocked cell-cycle progression in colon cancer cells by silencing PI3K/AKT signalling, accompanied by the expression of cycle-related proteins such as cyclin-dependent kinase (including CDK2, CDK4 and CDK6) and cyclins (Cyclin B1 or D1) [12,13,14]. In addition, the pharmacological inhibition of PI3K/AKT may promote colon cancer cell apoptosis by regulating apoptosis-related proteins such as PARP, Bcl2 and Bax, and show effective antitumor effects [12,15]. The effective mean to target PI3K/AKT is thus the prospect of developing colon cancer therapeutics.

Tea is a traditional functional beverage in which the active ingredients, such as catechins, tea polyphenols, caffeine, and flavonoids, are widely considered to be effective in inhibiting tumor cell proliferation and development [16]. For example, camellia ptilophylla extract significantly promoted colon cancer cell HCT116 apoptosis, which was attributed to a decrease in AKT phosphorylation [17]. However, different varieties of tea, fermentation time or time of aging (years of storage) may have great differences in the production of active ingredient contents and anti-cancer effects. Hakka stir-fried green tea (HSGT) is a traditional tea in Guangdong, China. HSGT is a type of roasted green tea based on the technique of high-temperature and long-time final roasting, and aged HSGT has demonstrated anti-sputum and anti-stasis effects [18,19]. Nonetheless, its potential role in anti-cancer treatment and the influence of different production processes have not been investigated systematically. Herein, we compared comprehensively the active components of HSGT aged for different lengths of time and studied their functions in the regulation of cell proliferation, cell cycle and apoptosis in colon cancer cells (HT-29). Our results showed for the first time that that traditional HSGT, after aging for years, markedly reduced the activity of PI3K/AKT signalling and its downstream target gene clusters from cell cycle or apoptosis.

## 2. Materials and Methods

### 2.1. Cells and Freeze-Dried Powder of Tea Extract

Colon cancer cell line HT-29 was purchased from the National Collection of Authenticated Cell Cultures of Chinese Academy of Sciences (Shanghai, China), and cultured in a medium containing 90% McCoy’s 5A Medium (Invitrogen, CA, USA) and 10% FBS (Gibco, CA, USA). The tea samples of Hakka stir-fried green tea (HSGT), from 2003 (**03Y**), 2007 (**07Y**), 2011 (**11Y**), 2015 (**15Y**) and 2019 (**19Y**), respectively, were purchased from Meizhou Junbao Industrial Co., Ltd (Meizhou, Guangdong province, China). A certain weight of HSGT was ground into crude powders and then mixed with water using a solid-to-liquid ratio of 1: 20 (gram: mL) at 90 °C. The tea soup was leached for 30 min and immediately filtered, and the above procedures were repeated 3 times. Then, the tea soup was combined and concentrated to 1/10 of the original volume in a steam bath. Finally, the concentrated solution was lyophilized into lyophilized powder.

### 2.2. Analysis of Active Ingredients in Tea Extract

The analytical method for the tea extracts was based on our previous reported study [20]. According to GB/T 8313-2018 and GB/T 8314-2013 standards, we determined the content of tea polyphenols and free amino acids in lyophilized HSGT powders. The content of soluble sugar and flavonoids was determined by the anthrone–sulfuric acid colorimetric method and the aluminum trichloride colorimetric method. The monomer components of catechin, gallic acid and caffeine were measured by a high-performance liquid chromatograph (Agilent 1200 Series, Palo Alto, CA, USA). The chromatographic column used was a phenomenex C18 column (150 × 4.6 mm, 5 µm), and the mobile phase included phase A (containing 0.5% acetic acid, 1% acetonitrile and 2% methanol) and phase B (containing 0.5% acetic acid, 10% acetonitrile and 20% methanol). In the first 30 min of elution, phase A decreased from 72.5% to 20%, while phase B increased from 27.5% to 80%. At 30-35 min of elution time, phase A increased from 20% to 72.5%, and phase B decreased from 80% to 27.5%, and then continued to 40 min. The injection volume was 10 μL. The flow rate was 1.0 mL/min. The temperature was maintained at 28 °C. All the major compounds (Table 1 and Table 2) found in the tea extracts were verified with their corresponding standards using HPLC. The qualitative analysis results are given in the Supporting Information. In addition, the quantitative analysis was performed by the external standard method according to the peak area at the wavelength of 280 nm.

### 2.3. Cell Viability Assay

The concentration of HT-29 cells in log phase was adjusted to 5 × 10^4^ cells/mL, and then added to 96-well plates at 100 μL/well. After 24 h, the culture medium of adherent HT-29 cells was discarded, and 100 μL/well of HSGT solutions of different concentrations (lyophilized powder dissolved in McCoy’s 5A Medium) were added, respectively. As a control well, McCoy’s 5A Medium without FBS was added at 100 μL/well. Each group was repeated 4–6 times and cultured at 37 °C 5% CO_2_ for 24 h or 48 h. At the time of detection, 10 μL/well MTT solution (Beijing MYM Biotechnology Co., Ltd., Beijing, China) was added and co-cultured for 3–4 h. Subsequently, the medium in each well was aspirated, and 150 μL/well of MDSO (Biosharp, Anhui, China) solution was added and incubated for 10 min. A TriStar LB941 multifunctional microplate reader (Berthold Technologies, Baden Württemberg, Germany) was used to measure OD value at a wavelength of 490 nm and obtained the cell viability.

### 2.4. Cell Cycle Assay

HT-29 cells were treated with 0.2 mg/mL HSGT extract solutions with different storage years for 24 h/48 h. Trypsinized cells were resuspended and centrifuged in 1 mL of ice-cold PBS. Then, 1 mL of 70% ethanol in an ice bath was added to the cell pellet, mixed by pipetting, and fixed for 12–24 h. After washing with ice PBS, cells were added to 0.5 mL propidium iodide (PI) staining solution (C1052 Cell Cycle and Apoptosis Detection Kit, Beyotime Biotechnology, Shanghai, China) for 30 min at 37 °C in the dark. The cell suspension was then filtered through a filter and placed on ice. A flow cytometer (Accuri C6 Plus, BD, Lake Franklin, NJ, USA) was used to detect the fluorescence signal at the excitation wavelength 488 nm channel. Flow cytometry data was processed using FlowJo-V10 software.

### 2.5. Cell Apoptosis Assay

HT-29 cells were treated with 0.2 mg/mL HSGT solution of different storage years for 24 h or 48 h, in which 5 μM cisplatin (MedChemExpress, Princeton, NJ, USA) was used as a positive control for apoptosis. A total of 1 × 10^5^ digested HT-29 cells were resuspended with 195 μL Annexin V-FITC binding solution (C1062L Annexin V-FITC Apoptosis Detection Kit, Beyotime Biotechnology), and then 5 μL fluorescein isothiocyanate (FITC) and 10 μL PI staining solution were added. The cell suspension was incubated at room temperature (20–25 °C) for 10–20 min in the dark, then filtered through a filter and placed on ice. Under the excitation wavelength of a 488 nm flow cytometer (Accuri C6 Plus, BD, Lake Franklin, NJ, USA), the FITC fluorescence signal at 515 nm and the PI fluorescence signal at 560 nm were detected. FlowJo-V10 software was used to process and analyze the data.

### 2.6. Western Blotting Assay

RIPA lysate (Beyotime Biotechnology) was used to extract proteins from HT-29 cells treated with HSGT extract solution. The protein concentration was determined with a BCA protein detection kit (Thermo Fisher Scientific, Waltham, MA, USA), followed by purified water and 4×loading buffer (Solarbio, Beijing, China) to adjust the protein concentration. Protein samples were separated by SDS-PAGE electrophoresis and transferred to PVDF membranes (Millipore, Boston, MA, USA). Then, 5% non-fat dry milk (Bio-FROXX, Einhausen, Germany) was used to block the PVDF membrane and then incubated with primary antibodies overnight at 4 °C. Primary antibodies incubated included: p-PI3K (Tyr458) antibody (CST 4228, Cell Signaling Technology, Boston, MA, USA), PI3K antibody (CST 4292), p-AKT (Ser473) antibody (CST 4060), AKT antibody (CST 4691), Bax antibody (Abcam ab32503, Cambridge, UK), Bcl-2 antibody (Abcam ab117115), PARP antibody (CST 9542), CDK2 antibody (CST 2546), CKD4 antibody (CST 12790S), Cyclin B1 (CST 4138) and β-actin antibody (CST 4970). The PVDF membrane was washed and incubated with secondary antibodies, including HRP-labeled goat anti-rabbit IgG antibody (KPL 074-1506, SeraCare Life Sciences, Milford, MA, USA) and HRP-labeled goat anti-mouse IgG antibody (KPL 074-1806, SeraCare Life Sciences) at room temperature for 50 min. Blot signals on membranes were imaged by a chemiluminescent gel imaging system (Tanon 5200, Shanghai, China). Band grayscale was analyzed with Image J software.

### 2.7. Data Statistics and Analysis

All data were presented as mean ± standard deviation (SD), and each experimental data was repeated at least three times under the same conditions. Data analysis was performed using SPSS 20.0 and GraphPad Prism 8.0 software. One-way ANOVA analysis was performed between the treatments of multiple tea samples. Different lowercase letters in the same row/group show the significant differences at *p* < 0.05 level.

## 3. Results

### 3.1. Identification of Active Ingredients in the Extracts of Chinese Green Tea (HSGT) Aged for Different Years

Active ingredients such as polyphenols, catechins and amino acids in tea are believed to be helpful for anti-cancer treatment. We first analyzed Chinese green tea (HSGT) aged for different years, including five batches that had been stored since 2003 (**03Y**), 2007 (**07Y**), 2011 (**11Y**), 2015 (**15Y**) and 2019 (**19Y**). The components of the main active ingredients obtained from the tea extract (lyophilized powders) were analysed with HPLC. As shown in Table 1, for the conventional composition of the teas, the content of polyphenols found in samples of **11Y** and **15Y** was significantly higher than that of other samples (*p* < 0.05). The amino acid and soluble sugar contents found in **03Y** and **19Y** were significantly higher than that of other samples (*p* < 0.05). For the ratio of phenol to ammonia and flavonoids, the lyophilized powder of **11Y** tea showed the highest content. For the analysis of catechins, gallic acid and caffeine, there were no significant differences in the content of total catechins, ester catechins and non-ester catechins between these aged teas (Table 2). However, the content of catechin (C), catechin gallate (CG), gallocatechin (GC), gallocatechin-3-gallate (GCG), gallic acid (GA), and caffeine (CAFF) in the freeze-dried powder of the extract of teas with long aging time (**03Y**: 18-year-aged, **07Y**: 14-year-aged) was found to be significantly increased compared with other groups (*p* < 0.05). Epigallocatechin (EGC) and epicatechin (EC) showed higher content in the short aging time (**11Y**: 10-year-aged, **15Y**: 6-year-aged and **19Y**: 2-year-aged) compared to **03Y** or **07Y** (*p* < 0.05). Epicatechin-3-gallate (ECG) showed the highest content in a 10-year aged tea (**11Y**). Moreover, there was no significant difference found for epigallocatechin-3-gallate (EGCG) among the groups. Taken together, the data obtained may reveal differentially active components in the teas aged for different years. Their therapeutic effect on colon cancer cells was investigated in detail in the following sections.

### 3.2. Aged Chinese Green Tea (HSGT) Inhibits the Proliferation of Colon Cancer Cells

We evaluated the effect of aged-tea extracts on the proliferation of HT-29 cancer cells. The HT-29 cells were treated with the solutions of aged-tea extracts at different concentrations for 24 and 48 h and followed by MTT assays to identify cell viability. The results showed that extracts at high concentration (1.0 mg/mL) significantly inhibited the viability of HT-29 cells at either 24 or 48 h of treatment compared to the lower concentration group (0.2 mg/mL) or the control group with buffer (*p* < 0.05). While treated for 24 h at each specific concentration, no significant effect on cell viability was observed for the tea aged for different years (Figure 1A). Nonetheless, the results obtained for 48 h treatment were obviously varied. As shown in Figure 1B, the **19Y** sample aged for 3 years generally showed lower cell viability than the tea samples of **03Y** and **07Y** at the treatment concentrations of 0.2 and 0.4 mg/mL, respectively (*p* < 0.05). These results also indicated that **19Y** significantly inhibited the viability of HT-29 cells in a concentration-dependent manner. We also found that the IC_50_ value of MTT in HT-29 cells had statistically significant correlations with the content of ECG and GA (positive correlation) and EC and EGC (negative correlation) in the tea extracts (*p* < 0.05) (Appendix A). Moreover, the results may suggest that the tea samples such as **15Y** and **19Y** with shorter aging time could be more advantageous in the inhibition of cell viability under 48 h treatment conditions.

### 3.3. Aged Chinese Green Tea (HSGT) Promotes Cell Apoptosis

We further investigated whether the teas with different aging time could increase apoptosis in HT-29 cells, which could be a possible factor causing the decrease of HT-29 cell viability. Since significant differences in cell viability inhibition were observed for the teas with different years of aging at 0.2 mg/mL for 48 h treatment (*p* < 0.05) (Figure 1B), we thus selected this concentration for subsequent experiments. For the HT-29 cells, after treatment with 0.2 mg/mL of different tea extracts for 24 h, it was found that the apoptosis rate of samples **03Y** and **19Y** was significantly increased compared with the control group (*p* < 0.05). It was noteworthy that sample **03Y** exhibited a very comparable effect to the positive control using cisplatin (Figure 2A,B). For the 48 h treatments, compared to the control, all the aged teas examined generally enhanced the apoptosis significantly in the HT-29 cells. Among these aged teas, sample **03Y** was found to have the highest apoptosis-promoting effect and it was also significantly higher than the positive control (*p* < 0.05) (Figure 2C,D). In addition, the cell apoptosis rate of HT-29 cells was positively correlated with the content of soluble sugar and GCG (*p* < 0.05) (Appendix A). These results may confirm that a significant enhancement in HT-29 cell apoptosis was induced by the aged teas tested. In particular, sample **03Y** was found to be the most potent one.

### 3.4. Aged Chinese Green Tea (HSGT) Slows Down Cell-Cycle Progression

We then examined whether the aged teas could regulate cell-cycle progression in HT-29 cells. In the assays, the tea extracts of **03Y**, **07Y**, **11Y**, **15Y**, **19Y** and control were investigated with HT-29 cells at a concentration of 0.2 mg/mL for 24- and 48-h, respectively. Cell-cycle analysis was then performed with flow cytometry for comparison (Figure 3A,C). Statistical results showed that all these aged teas were able to increase the proportion of G1 and S phases and decrease the proportion of G2/M phase in the HT-29 cell population, regardless of treatment time (*p* < 0.05) (Figure 3B,D). In particular, the ratio of G1 phase and S phase for the cells treated with sample **19Y** was significantly higher than that of other samples (*p* < 0.05) (Figure 3D). Moreover, the cell-cycle distribution of G1/G0 phase presented statistically significant correlations with the components of EC and EGC (positive correlation, *p* < 0.01), GC and GA (negative correlation, *p* < 0.01) and ECG and total catechins (negative correlation, *p* < 0.05) (Appendix A). Moreover, the cell-cycle distribution of S phase was negatively correlated with TP/FAA and C (*p* < 0.05) (Appendix A). Therefore, we may conclude that the aged teas could block the transition of the G1/S phase population of HT-29 cells to the G2/M phase, thereby reducing cell viability.

### 3.5. Aged Chinese Green Tea (HSGT) Reduces the Expression of Cell-Cycle-Related Proteins in Colon Cancer Cells

Due to the inhibitory role of the aged teas in cell-cycle progression, we examined further several major cell-cycle regulatory proteins. The levels of CDK2, CDK4 and CylinB1 in HT-29 cells treated with 0.2 mg/mL of the extract of the aged teas for 48 h were examined and compared with the Western blot results (Figure 4A). Quantitative analysis of protein bands revealed that the aged tea generally caused significant decreases in CDK2, CDK4 and CylinB1 protein levels (*p* < 0.05) (Figure 4B–D). The results indicate that aged tea may effectively block cell-cycle progression by inhibiting the expression of CDK2, CDK4 and CylinB1 proteins.

### 3.6. Aged Chinese Green Tea (HSGT) Inactivates PI3K/AKT Signalling and Enhances Apoptotic Pathways

Activation of PI3K/AKT signalling is essential for cancer cell viability, including colon cancer, and also maintains downstream inhibitory signals of apoptosis. We therefore examined the levels of phosphorylated PI3K (p-PI3K), phosphorylated AKT (p-AKT), PRAP, Bax and Bcl2 in HT-29 cells treated with 0.2 mg/mL extract of HSGT for 48 h by Western blotting (Figure 5A). Quantitative analysis of protein bands revealed that the expression of p-PI3K and p-AKT in HT-29 cells were markedly reduced after the treatment with the extracts of aged teas, demonstrating an inhibitory effect on PI3K/AKT signalling (*p* < 0.05) (Figure 5B,C). For the apoptotic pathway, the tea extracts increased PRAP expression and the Bax/Bcl2 ratio in HT-29 cells (*p* < 0.05) (Figure 5D,E). In particular, for the cells treated with the tea extracts of **15Y** and **19Y**, p-AKT expressions were significantly lower than that of other tea samples (*p* < 0.05) (Figure 5C). In addition, their Bax/Bcl2 ratio was also significantly higher than others (*p* < 0.05) (Figure 5E). These results demonstrate the inhibitory effect of the tea extracts on PI3K/AKT signalling and a sustained enhancement in the apoptotic pathway.

## 4. Discussion

As one of the most severe and refractory cancer types, the pathogenesis of colon cancer and its effective treatment options are constantly being explored. It is generally believed that the pathogenesis of colon cancer is related to the accumulation of gene mutations in colon epithelial cells and disturbance of the immune microenvironment [21,22]. In response to these characteristics, advanced drug treatment options, including immunotherapy (anti-PD1 or anti-CTLA4 therapy) and targeted drugs (such as bevacizumab and cetuximab), have already been used in the treatment of patients or undergone clinical trials [23,24]. However, these drugs still cannot avoid tumor recurrence and the side effects lead to a serious decline in the life quality of patients. The main purpose of the present study is to search for active and low-toxicity natural substances from traditional Chinese herbs, such as aged Chinese green tea (HSGT), to ensure both efficacy and safety against colon cancer.

Our study identified the main components from the extracts of the HSGT aged with different years. The in vitro activity of the aged-tea extracts against the proliferation of colon cancer cells was systematically investigated and compared for the first time. In fact, previous studies have confirmed the beneficial effects of different kinds of tea extracts in colon cancer treatment [25,26,27]. In a study of Japanese men, it was reported that green tea may possibly reduce the risk of colon cancer [25]. Among the bioactive components studied in green tea, EGCG down-regulated STAT3 expression to induce apoptosis in colon cancer cells (SW480) [26]. In addition, catechins (mainly EGC and EGCG) were able to restrict the proliferation of HCT116 colon cancer cells [27]. These results were also validated in the present study, and we found that the tea samples of **15Y** and **19Y** showed the greatest inhibitory effect on cell viability among different aged teas. The HPLC analysis confirmed that the extracts of **15Y** and **19Y** contained high levels of EGC and EC. Given that there was no significant difference found in the EGCG content across different aging periods in our results, we thus reasoned that the effective anticancer activity against HT-29 could be probably mediated by the EGC and EC contents of the aged teas.

A previous study found that HSGT potentially activated the AMPK cascade signalling against liver injury in a high-fat-diet-induced obesity model [20]. However, the therapeutic potential of the aged HSGT in anticancer was not investigated. In the present study, we provided evidence that these aged teas in colon cancer therapy exhibited biological growth inhibition of HT-29 cells regardless the time of aging. However, for apoptosis and cell-cycle regulation, there was variability in the tea aged for different years. For example, with an effective treatment time of 48 h, among these aged teas, sample **03Y** induced the most apoptosis and **19Y** had the most severe arrest in the G1/S phase of the cell cycle. The results may indicate that the time of HSGT aging is important in improving their potency in anticancer treatment. Moreover, many Chinese teas, including Keemun black tea, Qingzhuan tea and Pu-erh tea had different active ingredients depending on the number of years they were aged [28,29,30,31,32]. The changes of active components in tea due to aging time may result in significant differences in cancer therapy or treatment for other diseases. For example, the content of statins and polyphenols in Pu-erh tea fermented for a short period of 42 days was higher than that with longer fermentation time, which is beneficial for relieving cardiovascular disease caused by hyperlipidemia [33]. Given the different apoptosis and cell-cycle regulation of the teas with different aging time in our study, the aged teas combined with cyclin inhibitors or apoptosis-promoting molecules can be utilized to develop an effective management strategy for colon cancer in different scenarios.

For the aged teas of **03Y** and **19Y**, both contained amino acid and soluble sugar contents that were significantly higher than the other samples tested. For the difference of the **03Y** and **19Y** samples, GA in the **03Y** group was found to be significantly higher than that in the **19Y** group, while EGC was higher in the **19Y** group. We therefore hypothesized that GA in **03Y** group mediated apoptosis, while EGC in **19Y** group might affect cell-cycle regulation. It was reported that GA induced the apoptosis of HCT-15 colon cancer cells in a ROS-dependent manner [34]. Furthermore, the natural plant-derived GA inhibited PI3K/AKT phosphorylation and increased colon cancer cell apoptosis [35]. In terms of cell-cycle regulation, it was previously found that EGC kept Lovo colon cancer cells in the G1 phase [36] and inhibited the expression of cyclin D1 and CDK4 in tumor cells [37]. Thus, EGC is also believed to be involved in the activation of p-PI3K- and p-AKT-induced tumor growth signals [38]. Taken together, all these results may point to the fact that the active ingredients in the aged teas activate PI3K/AKT signalling to achieve the inhibition of colon cancer cell growth.

## 5. Conclusions

In summary, Chinese green teas (HSGT) aged for different years show certain effects in the inhibition of colon cancer cell growth, arrests in the G1/S phase of the cell cycle, and induce apoptosis, most likely via suppressing the PI3K/AKT signalling pathway, downregulating CDK2, CDK4 and CylinB1 proteins, and upregulating PARP protein and the ratio of Bax/Bcl-2.

## Figures and Tables

**Figure 1 molecules-27-08272-f001:**
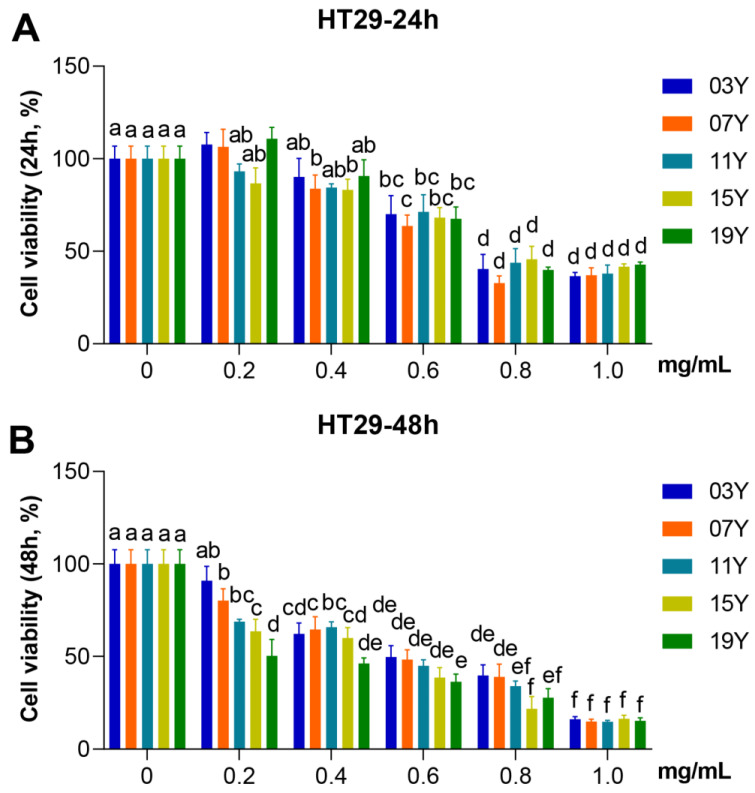
Inhibitory activity of HSGT extracts with various storage years on HT-29 cell viability. HT-29 cells were treated with 0, 0.2, 0.4, 0.6, 0.8 and 1.0 mg/mL extracts of the different tea samples **03Y**, **07Y**, **11Y**, **15Y** and **19Y** for 24 (**A**) and 48 h (**B**), respectively. The cell viability was detected by MTT assay. One-way ANOVA analysis was performed between the treatments of multiple tea samples. The different lowercase letters (a, b, c, d, e, f) in the same group show the significant differences at *p* < 0.05 level.

**Figure 2 molecules-27-08272-f002:**
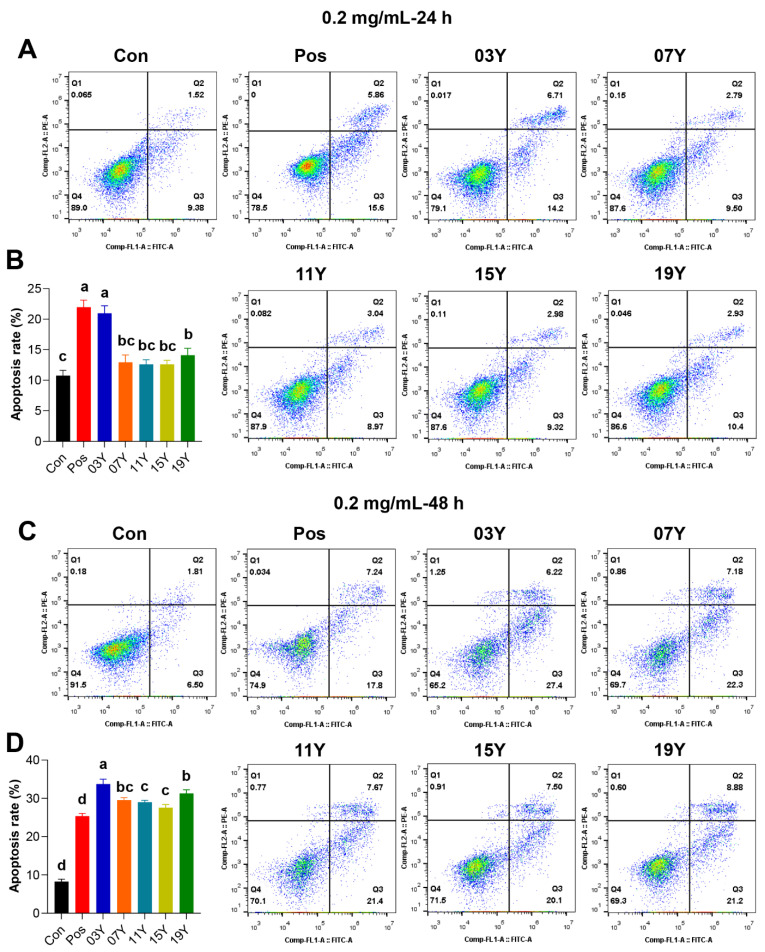
The promotion of apoptosis of HT-29 cells by tea extracts with various storage years. HT-29 cells were treated with 0.2 mg/mL extract of different tea samples **03Y**, **07Y**, **11Y**, **15Y** and **19Y** for 24 (**A**,**B**) and 48 h (**C**,**D**), respectively. Total 0.2 mg/mL cisplatin was used as a positive control (Pos group). The proportion of apoptotic cells in each group was examined and counted using flow cytometry. One-way ANOVA analysis was performed between the treatments of multiple tea samples. The different lowercase letters (a, b, c, d) in the same group show the significant differences at *p* < 0.05 level.

**Figure 3 molecules-27-08272-f003:**
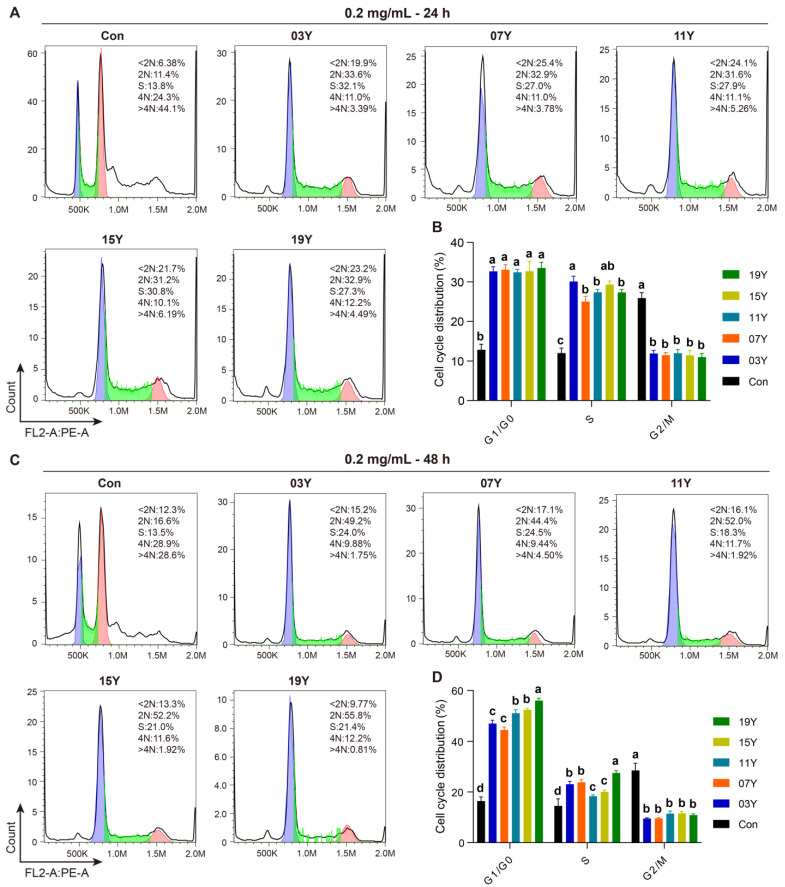
The tea extract inhibited HT-29 cell-cycle progression. HT-29 cells were treated with 0.2 mg/mL extracts of different tea samples **03Y**, **07Y**, **11Y**, **15Y** and **19Y** for 24 and 48 h, respectively (**A**,**C**). The phase ratios of total cell cycle in each group were examined using flow cytometry (**B**,**D**). One-way ANOVA analysis was performed between the treatments of multiple tea samples. The different lowercase letters (a, b, c, d) in the same group show the significant differences at *p* < 0.05 level.

**Figure 4 molecules-27-08272-f004:**
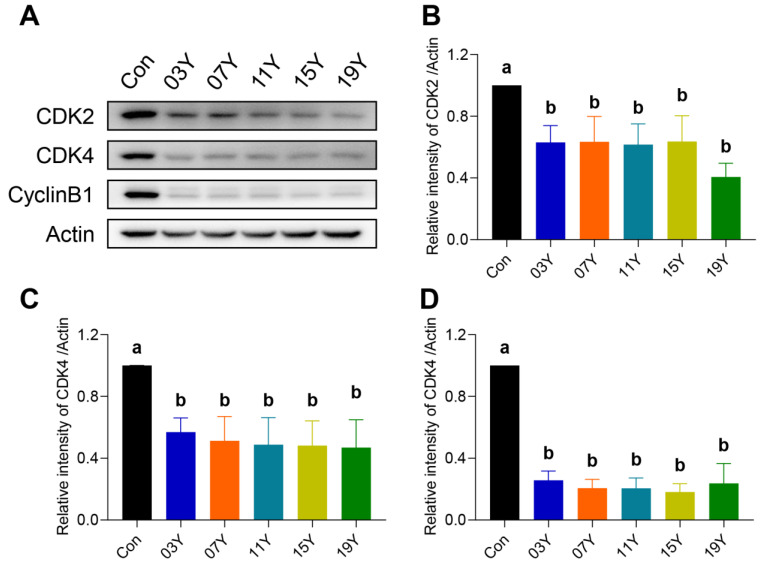
Th tea extract inhibited the expression of cell-cycle-related proteins. (**A**). HT-29 cells were treated with 0.2 mg/mL extract of different tea samples **03Y**, **07Y**, **11Y**, **15Y** and **19Y** for 48 h, and then CDK2, CDK4 and CylinB1 expressions were detected by Western blotting. (**B**–**D**). Relative quantification of CDK2, CDK4 and CylinB1 protein levels in each group was shown, with actin as a reference. One-way ANOVA analysis was performed between the treatments of multiple tea samples. The different lowercase letters (a and b) in the same group show the significant differences at *p* < 0.05 level.

**Figure 5 molecules-27-08272-f005:**
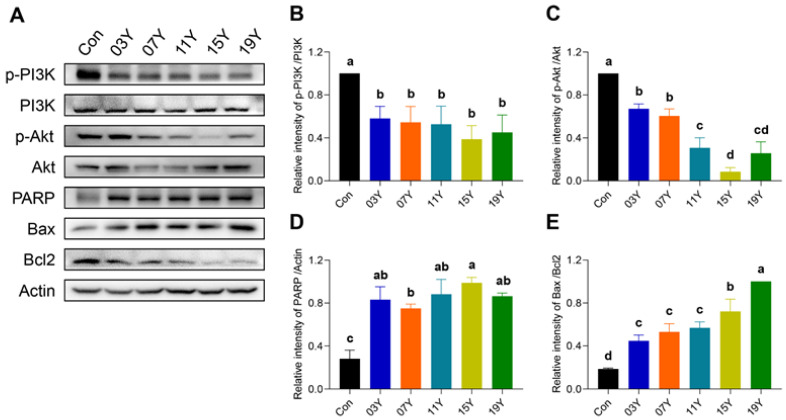
The tea extract inactivated PI3K/AKT signalling and enhanced apoptotic pathways. (**A**). HT-29 cells were treated with 0.2 mg/mL extracts of different tea samples **03Y**, **07Y**, **11Y**, **15Y** and **19Y** for 48 h, followed by detection of p-PI3K, p-AKT, PRAP, Bax and Bcl2 expression by Western blotting. (**B**–**E**) Relative quantification of p-PI3K, p-AKT, and PRAP protein levels (with actin as references) and Bax/Bcl2 ratio in each group was shown. One-way ANOVA analysis was performed between the treatments of multiple tea samples. The different lowercase letters (a, b, c, d) in the same group show the significant differences at *p* < 0.05 level.

**Table 1 molecules-27-08272-t001:** Analysis of conventional components from the freeze-dried powder extracted from the aged Chinese green tea (HSGT) ^a^.

Ingredients	03Y	07Y	11Y	15Y	19Y
Tea polyphenols/%	39.26 ± 0.24 b	32.39 ± 0.33 c	41.65 ± 0.09 a	39.91 ± 0.22 ab	38.27 ± 0.01 b
Free amino acid/%	5.35 ± 0.04 ab	4.40 ± 0.00 bc	4.19 ± 0.02 c	5.01 ± 0.08 b	5.51 ± 0.09 a
Ratio of TP to FAA	7.44 ± 0.06 c	7.38 ± 0.30 c	9.93 ± 0.05 a	8.14 ± 0.06 b	7.04 ± 0.01 c
Soluble sugar/%	26.84 ± 0.05 a	23.69 ± 0.62 b	24.34 ± 0.53 b	23.64 ± 0.26 b	26.65 ± 0.29 a
Flavonoids/%	2.60 ± 0.04 d	3.01 ± 0.03 b	3.25 ± 0.05 a	2.85 ± 0.03 c	2.59 ± 0.01 d

^a^ Values represent means ± SD (n = 4). One-way ANOVA analysis was performed between the treatments of multiple tea samples. The different lowercase letters (a, b, c, d) in the same group show the significant differences at *p* < 0.05 level. TP, tea polyphenols; FAA, free amino acid; five aged HSGT samples, **03Y**, **07Y**, **11Y**, **15Y**, and **19Y** refer to the year of storage in 2003, 2007, 2011, 2015, and 2019, respectively.

**Table 2 molecules-27-08272-t002:** Quantitative analysis of catechins, gallic acids and caffeines in the extract of the aged Chinese green tea (HSGT) a.

Ingredients (mg/g)	03Y	07Y	11Y	15Y	19Y
CG	2.64 ± 0.05 a	1.67 ± 0.04 ab	1.65 ± 0.14 ab	0.76 ± 0.04 b	1.08 ± 0.08 ab
ECG	29.73 ± 0.96 b	33.74 ± 0.37 ab	33.94 ± 0.81 a	30.34 ± 1.18 b	26.51 ± 0.21 b
GCG	16.47 ± 0.39 a	10.45 ± 0.48 b	11.44 ± 0.21 b	11.13 ± 0.51 b	12.24 ± 0.24 b
EGCG	137.55 ± 1.04 a	141.79 ± 0.04 a	146.77 ± 0.85 a	131.21 ± 0.98 a	131.20 ± 0.63 a
EC	9.65 ± 0.11 b	9.31 ±0.25 b	12.41 ± 0.21 a	13.52 ± 0.47 a	14.37 ± 0.15 a
C	6.17 ± 0.03 a	4.48 ± 0.13 ab	6.49 ± 0.02 a	5.46 ± 0.14 ab	4.35 ± 0.26 b
GC	69.02 ± 0.00 b	86.00 ± 1.28 a	51.71 ± 0.85 c	52.05 ± 1.87 c	44.13 ± 0.73 c
EGC	38.65 ± 1.25 c	33.02 ± 0.77 c	45.03 ± 0.09 b	58.59 ± 1.82 a	63.67 ± 0.36 a
GA	10.88 ± 0.03 a	9.79 ± 0.17 b	6.40 ± 0.22 c	5.54 ± 0.33 c	3.45 ± 0.03 d
CAFF	92.67 ± 0.25 a	90.45 ± 1.47 a	75.85 ± 0.10 b	88.99 ± 1.06 ab	71.96 ± 0.01 b
Ester catechins	185.59 ± 4.96 a	184.01 ± 2.66 a	180.13 ± 7.42 a	188.57 ± 7.20 a	169.85 ± 2.02 a
Non-ester catechins	120.91 ± 2.13 a	132.37 ± 0.72 a	115.26 ± 0.15 a	129.62 ± 4.30 a	128.21 ± 2.61 a
Total catechins	306.50 ± 5.45 a	328.23 ± 2.40 a	308.18 ± 15.38 a	303.29 ± 11.49 a	298.06 ± 4.89 a

^a^ Values represent means ± SD (n = 4). One-way ANOVA analysis was performed between the treatments of multiple tea samples. The different lowercase letters (a, b, c) in the same group show the significant differences at *p* < 0.05 level. CG, catechin gallate; ECG, Epicatechin-3-gallate; GCG, gallocatechin-3-gallate; EGCG, epigallocatechin-3-gallate; EC, epicatechin; C, catechin; GC, gallocatechin; EGC, epigallocatechin; GA, gallic acid; CAFF, caffeine.

## Data Availability

The data presented in this study are available in Appendix A.

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
