# Peer review of "Induction of Apoptosis via Inactivating PI3K/AKT Pathway in Colorectal Cancer Cells Using Aged Chinese Hakka Stir-Fried Green Tea Extract"

_molecules, 2022, doi:10.3390/molecules27238272_

Round 1

Reviewer 1 Report

The paper presents an interesting study of employing active and low toxic natural substances from traditional Chinese herbs, such as the aged Chinese green tea (HSGT), to ensure both the efficacy and safety against colon cancer. The results and discussion are presented well and I recomend publication of the manuscript after minor revision. Authors should crosscheck whole manuscript for enlish spell check and some gramatical errors highlighted in the reviewer comments given in the attached PDF. Good Luck for publication.

Author Response

Please refer the attached file. Thank you

Reviewer 2 Report

In this manuscript, the effects that traditional Chinese aged tea (HSGT) may inhibit colon cancer cell proliferation, cell cycle progression and pro-moted apoptosis of colon cancer cells by inactivating PI3K/AKT signalling were studied by Wong, Wen and co-workers. Based on the experiment result, they said the active ingredients in the aged teas activates PI3K/AKT signalling to achieve the inhibition of colon cancer cell growth, down-regulating CDK2, CDK4 and CylinB1 proteins, and upregulating PARP protein and the ratio of Bax/Bcl-2. However, I think the authors should provide more information in this paper.

The current data showed irrelevant relationship for the tea samples between the concentrations of the ingredient and the ages. 03Y has higher GA, 15Y and 19Y have higher EGC. The authors should provide more information to prove the concentration change is because of the age difference rather than the sample bias.

Round 2

Reviewer 2 Report

I think the revised version is good, I agree this paper can be accepted.